# Automated optimized parameters for T-distributed stochastic neighbor embedding improve visualization and analysis of large datasets

Anna C. Belkina [1,2]*, Christopher O. Ciccolella[3], Rina Anno[4], Richard Halpert [5], Josef Spidlen [5] & Jennifer E. Snyder-Cappione[2,6]

Accurate and comprehensive extraction of information from high-dimensional single cell datasets necessitates faithful visualizations to assess biological populations. A state-of-the-art algorithm for non-linear dimension reduction, t-SNE, requires multiple heuristics and fails to produce clear representations of datasets when millions of cells are projected. We develop opt-SNE, an automated toolkit for t-SNE parameter selection that utilizes Kullback-Leibler divergence evaluation in real time to tailor the early exaggeration and overall number of gradient descent iterations in a dataset-specific manner. The precise calibration of early exaggeration together with opt-SNE adjustment of gradient descent learning rate dramatically improves computation time and enables high-quality visualization of large cytometry and transcriptomics datasets, overcoming limitations of analysis tools with hard-coded parameters that often produce poorly resolved or misleading maps of fluorescent and mass cytometry data. In summary, opt-SNE enables superior data resolution in t-SNE space and thereby more accurate data interpretation.

[1] Department of Pathology and Laboratory Medicine, Boston University School of Medicine, Boston, MA 02118, USA. [2] Flow Cytometry Core Facility, Boston University School of Medicine, Boston, MA 02118, USA. [3] Omiq, Inc, Santa Clara, CA 95050, USA. [4] Department of Mathematics, Kansas State University, Manhattan, KS 66506, USA. [5] BD Life Sciences–FlowJo, Ashland, OR 97520, USA. [6] Department of Microbiology, Boston University School of Medicine, Boston, MA 02118, USA. *email: BELKINA@BU.EDU

Visual exploration of high-dimensional data is imperative for the comprehensive analysis of single cell datasets. Fluorescence, mass and sequencing-based cytometric data analysis requires tools that are able to reveal the combinations of proteomic and/or transcriptomic markers that define complex and diverse cell phenotypes in a mixed population. While traditional biaxial data presentation via expert-driven gating is still the standard analysis method for cytometry data, with the advent of the modern multi-parameter era an analysis tool that can accurately and comprehensively visualize multi-dimensional data is direly needed to relieve the current cytometry data-processing bottleneck.

To date, multiple dimensionality reduction techniques have been applied to cytometry data with variable success. Linear methods, such as PCA, are mostly unsuitable for cytometry data visualization as such techniques cannot faithfully present the non-linear relationships. t-Distributed Stochastic Neighbor Embedding (t-SNE) is a state-of-the-art dimensionality reduction algorithm for non-linear data representation that creates a low-dimensional distribution, or a 'map', of high-dimensional data[1,2]. Conspicuous groupings of datapoints, or 'islands', correspond to observations that are similar in the original high-dimensional space and help to visualize the general structure and heterogeneity of a dataset. When t-SNE embeds single cell data, the islands represent cells with similar phenotypes, as defined by a cytometric or genomic signature, thereby allowing to reveal biological data structure and to surface important differences between samples and/or subject groups[3]. In addition, t-SNE maps are used to categorize single cell data into relevant biological populations for downstream quantification, achievable through expert-guided filtering (gating)[4] or unsupervised clustering of the map[5–7]. Visualizations of cytometry data produced with other non-linear embedding algorithms, such as LargeVis[8], UMAP[9], and EmbedSOM[10], can be interpreted and interrogated in a similar manner. Clustering algorithms that directly interrogate high-dimensional data, such as FlowSOM[11] and PhenoGraph[12], are often used in conjunction with 2D maps to present annotated cell clusters to the viewer.

A limitation of t-SNE in its current form is its inability to scale to datasets with large numbers of observations[7,8]. This restrains t-SNE's utility for cytometry datasets that often include millions of observations (events) routinely collected for phenotypic analysis. Unlike PCA, t-SNE learns the embedding non-parametrically, and hence new pieces of data cannot validly be added to an existing analysis, necessitating the whole dataset to be analyzed within one computation. When the full dataset is comprised of multiple samples, each representing a subject in a large cohort or an independent experimental condition, retaining statistically significant representation of small subpopulations in each sample requires inflating the dataset size[13]. However, even within a single measurement, subsampling the data risks preventing rare subsets from being identified. These limitations cannot be overcome via application of the currently understood best practices for t-SNE use. Not only are large datasets computationally expensive to analyze, but also the resulting t-SNE maps provide poor visualization and incomprehensive representation of high-dimensional data. Consequently, researchers often resort to either subsampling their data to the very limit of detection of rare populations[14] or to exporting specific populations from their dataset, thus compromising the unbiased data analysis approach[15,16].

Although t-SNE has been widely adopted by the scientific community, to our knowledge no rigorous theoretical or empirical testing of t-SNE for cytometry applications has been performed. In 2013, Amir et al. first reported the use of Barnes-Hut (BH) implementation of t-SNE (or viSNE, as it was renamed[3]) on mass cytometry data; since then, BH-tSNE has

been integrated into the majority of commercial and open-source platforms for cytometry analysis. In most implementations, few or no adjustments were made to the Barnes-Hut t-SNE algorithm for the requirements of cytometry datasets; the default and hard-coded parameter settings that were originally tested and optimized with non-cytometry datasets like CIFAR (image dataset) or MNIST (handwritten digits) are retained in these cytometry programs and hence are still widely used. Recently published modifications of t-SNE, such as HSNE[17] and FIt-SNE[18], do not fully address these limitations. Therefore, development of rigorous methodology to release the full potential of t-SNE for single cell data comprehension and provide clues for optimization of second generation t-SNE-like algorithms is the primary motivation for this work.

In this study, we first assess the behavior of t-SNE computation with routinely used settings that match commonly prescribed best practices, and then iteratively modify parameters of embedding to identify conditions that ensure improved visualization. As a result, we propose a new method to automatically find dataset-tailored t-SNE parameters via fine-tuning of the early exaggeration phase of t-SNE embedding in real time. We call our approach opt-SNE, for optimized t-SNE. opt-SNE adjustments can greatly shorten the number of iterations required to obtain visualizations of large cytometry datasets with superior quality. Our approach also eliminates the need for trial-and-error runs intended to empirically find the most favorable selection of t-SNE parameters, potentially saving many hours of computation time per research project. Finally, we implemented opt-SNE as an open source fast multicore t-SNE C++ package which allows much faster computation of t-SNE embeddings than original single-core Barnes-Hut t-SNE implementation.

## Results

**The standard t-SNE fails to visualize large datasets.** The t-SNE algorithm can be guided by a set of parameters that finely adjust multiple aspects of the t-SNE run[19]. However, cytometry data analysis software often locks or severely restrains the tunability of those parameters, likely to provide a simplified, 'one-size-fits-all' solution for t-SNE use in the software packages. Although each software platform has a unique combination of possible adjustments, most allow changes to both the number of iterations and to the perplexity (a soft measure for the number of nearest neighbors considered for each data point).

The datasets used throughout this study include at least 1 million datapoints of fluorescent or mass cytometry data and are therefore considerably larger than the typical ($<5 \times 10^5$) datasets previously reported in benchmark studies of cytometry algorithmic tools[20]. Cytometry datasets larger than approximately $5 \times 10^5$ events are generally observed to produce poor quality t-SNE maps and are therefore usually subsampled prior to analysis.

Empirically, analysts have observed that increasing the number of iterations of t-SNE computation results in better quality maps[9,21]. We hypothesized that the resolution of t-SNE maps created from higher event counts could be dramatically improved via fine-tuning of t-SNE parameters. We first directly tested the relationship between iteration number and map quality by running two datasets (mass41parameter and flow18parameter, as described in Table 1) at the standard 1000 iterations per run and with an extended 3000 iteration computation (Fig. 1). As expected, 1000-iteration runs produced maps with poor visualization compared to 3000-iteration runs. Specifically, massive overlaps and random fragmentation of populations were observed. In contrast, the 3000-iteration runs resulted in maps with defined islands comprised of clearly isolated populations and no random fragmentations (Fig. 1a, b) The 1NN accuracy of

**Table 1 Datasets used in this paper**

| Dataset | Data type | Details | References |
|---|---|---|---|
| Mass41parameter | Mass cytometry | 41 parameter dataset (14 lineage parameters used for embedding) of 1 million datapoints concatenated from 5 samples of human bone marrow cells | 46 |
| Flow18parameter | Flow cytometry | 18 parameter dataset (11 lineage parameters used for embedding) of 1 million datapoints concatenated from 2 samples of human PBMC | 25,47 |
| Flow20M | Flow cytometry | 18 parameter dataset (15 lineage parameters used for embedding) of 20 million datapoints concatenated from 27 samples of human PBMC | 25,47 |
| 10X Genomics | scRNA-seq | Single cell gene expression data of E18 mouse brain pre-processed into 20 PCA projections used for t-SNE embedding | https://support.10xgenomics.com/single-cell-gene-expression/datasets and ref. 26 |
| van Unen et al. | Mass cytometry | 32 parameter dataset (26 lineage parameters used for embedding) of 5.22 million datapoints concatenated from 104 samples of PBMC and gut biopsy cells | 7 |

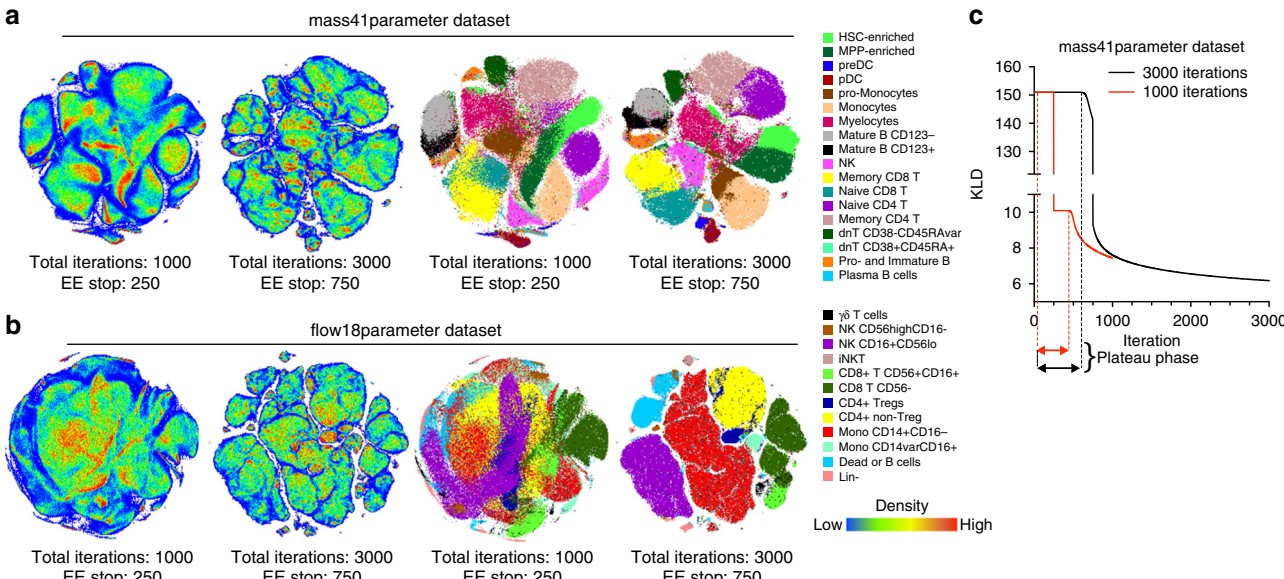

**Fig. 1** Performance of Barnes-Hut t-SNE implementation for cytometry data visualization. Standard (1000 iterations) and extended (3000 iterations) embeddings of mass cytometry (**a**) or flow cytometry (**b**) data are presented as heatmap density plots (left) or color-coded population overlays based on ground-truth classification of single cell in the datasets (right). **c** KLD change over iteration time of gradient descent for standard 1000 iterations (red line) or extended 3000 iterations (black line) embeddings of mass41parmeter dataset. Representative examples of multiple runs with varying seed values are shown

embedding was also higher in 3000-iteration embeddings for both datasets compared to the standard 1000-iteration t-SNE (Suppl. Fig. 1). Therefore, these findings are in agreement with the concept of a higher number of iterations resulting in higher quality t-SNE maps.

**KLD plateau phase resolves cluster structure in t-SNE.** In order to determine the cause of the difference in cluster resolution between the "standard" and "extended" t-SNE runs, we examined the behavior of KLD (Kullback-Leibler divergence, see Methods) over the duration of t-SNE embeddings (Fig. 1c). As expected, the KLD value was inflated during the EE since EEF is factored into gradient and KLD value calculation in BH-tSNE[1]. We applied EE over 250 iterations in the standard t-SNE configuration and over 750 iterations in the extended run with 3000 iterations since most platforms have EE scaled to 25% of total iteration number. Notably, the KLD did not immediately start to minimize at the beginning of the EE; instead, the graph of KLD over time is a plateau that is followed by a curve that captures the

incremental decrease of KLD, indicating the gradient descent. In the standard t-SNE run, the plateau was interrupted when the EE was stopped, then continued with a non-exaggerated value of KLD.

Van der Maaten and Hinton called EE a "trick" that improved resolution of the global structure of the data visualization that would not otherwise converge to separated clusters[1]. As the suboptimal quality of the 1000-iteration t-SNE maps shown in Fig. 1a, b demonstrated poor global structure resolution, we hypothesized that increasing the total iteration number in conventional cytometry analysis platforms would inadvertently increase the number of EE iterations resulting in improvement in map visualization. To test this hypothesis, we compared runs that differed in timing of the EE stop by plotting the embedding at the iteration when the EE stops and also at later iterations, thus assessing the effects of EE and our perturbations on both flow cytometry (Fig. 2a) and mass cytometry (Fig. 2b) data visualization.

We found notable differences in map quality between the runs with shorter and longer EE. Although the map after

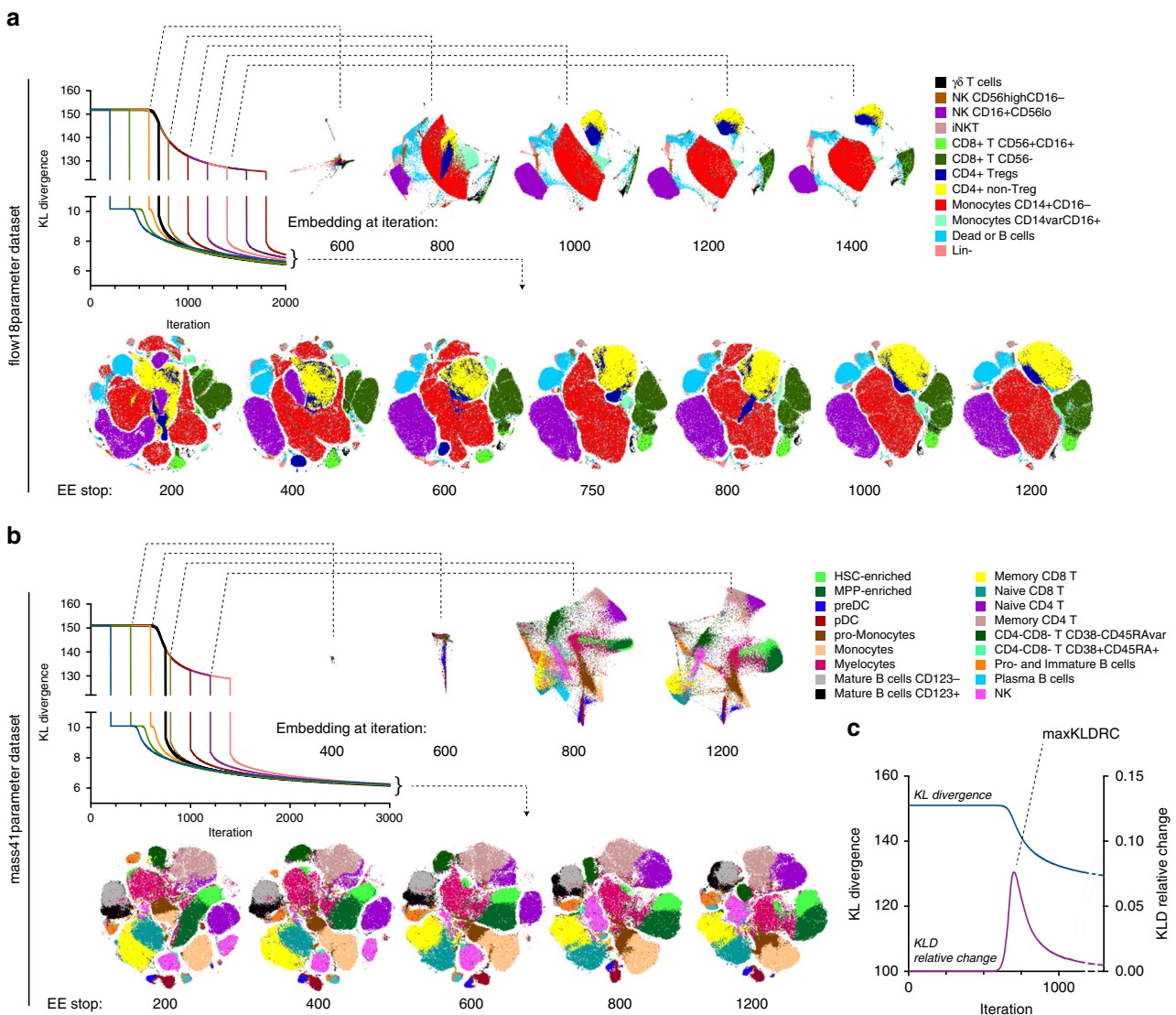

**Fig. 2** Effect of EE plateau phase on t-SNE visualization. EE was stopped after varying number of iterations and embedding visualization was examined at several intermediate timepoints and in the end of embedding for flow cytometry (total of 2000 iterations), (**a**) and mass cytometry (total of 3000 iterations). **b** Graphs showing KLD change over iteration time are color-labeled to distinguish curves corresponding to experiment perturbations, with black line indicating the run with the shortest EE but uninterrupted plateau. t-SNE maps are annotated with color-coded population overlays based on ground-truth classification of single cell in the datasets. **c** KLD and KLD relative change plotted against iteration time for the mass41parameter embedding. All embeddings were generated with standard BH-tSNE implementation and representative examples of multiple runs with varying seed values are shown

EE200/total3000 iterations may appear visually more appealing than previously shown EE250/total1000 map, ground-truth labeling indicated that phenotype-defined clusters are fragmented in both flow and mass cytometry maps. When cluster fragments were plotted on a biaxial plot against parameters that were used in the t-SNE dimension reduction, we were not able to identify protein marker expression patterns that immediately contributed to their fragmentation (Suppl. Fig. 2). Conversely, tight clusters that form at the end of the plateau remain mostly unchanged as long as the EE is being applied to the computation (Fig. 2a, b). The KLD minimization in that case could be explained by the gradual shrinking of the 2D space. Once EE is removed, the attractive forces within each cluster are weakened and the local structure of the data is fully resolved within each cluster.

Overall, our observations suggest that the EE phase of the gradient descent is essential for data clustering while the non-exaggerated descent results in the resolution of local structures.

When the EE is too short, the cell clusters continue to be resolved simultaneously with the local structure of each cluster being unfolded, leading to fragmented, overlapped or deformed islands in the resulting map (Fig. 2, Suppl. Fig. 2). To locate the timepoint when to end EE once the plateau phase is completed, we tracked the relative rate of KLD change ($KLDRC_N = 100\% \times ((KLD_{N-1} - KLD_N) / KLD_{N-1})$ where $N$ is the iteration number) and then identified the local maximum of KLDRC (maxKLDRC) (Fig. 2c). Since KLD is computed at each iteration, the maxKLDRC sensor can be added to the algorithm programmatically and would stop EE at the next iteration past maxKLDRC. For both datasets the maxKLDRC was detected at about 700–720 iterations (Fig. 2c), and, when the EE phase was followed by 300–1300 iterations without EE, the resulting visualization was similar to the EE750/3000 map at Fig. 1a and no cluster fragmentations were observed (Suppl. Fig. 3), demonstrating more relevant visualization of cytometry data compared to standard t-SNE settings.

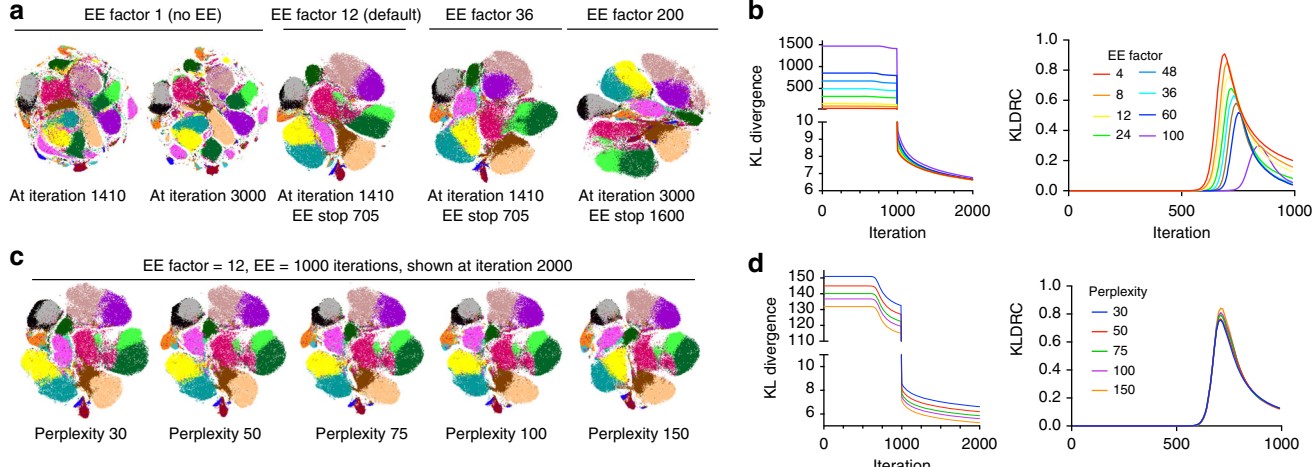

**Fig. 3** Effects of perplexity and EE factor adjustments on t-SNE visualization of cytometry data. **a**, **b** KLD, KLDRC, and t-SNE biaxial plots generated with varying EE factor values. **c**, **d** KLD, KLDRC, and t-SNE biaxial plots generated with varying perplexity. Graphs showing KLD and KLDRC change over iteration time are color-labeled to distinguish curves corresponding to experiment perturbations. Color overlays on t-SNE plots correspond to cell type classes labeled as in Figs. 1, 2. Representative examples of multiple runs with varying seed values are shown

**EE factor, perplexity tuning does not impact visualization**. Once we found EE to be crucial for map optimization, we next examined if the value of the EE factor $\alpha$ (EEF) can also be tuned to improve the results of t-SNE. We made $\alpha$ user-accessible in our C++ t-SNE code since it is hard-coded in the original Barnes-Hut C++ t-SNE implementation and all aforementioned results were obtained with default value of $\alpha = 12$. First, we tested how the optimization would proceed without EE ($\alpha = 1$). We expected the run to fail or produce extremely crowded results as explained in the original t-SNE report;[1] however, we did not see much overlap in cluster positioning, likely due to the substantial number of map iterations run (Fig. 3a). Nevertheless, the resulting map showed considerable fragmentation, reflecting an extreme case of an interrupted plateau phase. Even when run for as many as 3000 iterations, the fragmentation could not be remedied, again demonstrating the necessity of EE. As expected, higher values of $\alpha$ lead to much higher KLD during EE, however, the KLD values were similar at 2000 iterations when $\alpha$ varied between 4 and 60 (Fig. 3b) and EE was stopped at the max KLDRC iteration. Larger $\alpha$ prolonged the plateau phase and became detrimental for KLD values when over 100. Visually $\alpha = 200$ resulted in a distorted map with smaller populations lost. Therefore, we suggest that for cytometry applications the $\alpha$ parameter may remain unchanged and set to 12, as hard-coded in BH-tSNE[2], or reverted to $\alpha = 4$, as originally proposed by van der Maaten and Hinton[1] since per our results any value between 4 and 20 leads to similar outcomes.

Increased perplexity has been proposed to be an intuitively beneficial method for visualization improvement since it translates to a larger number of considered nearest neighbors and hence a more accurate approximation of attractive forces, while decreased perplexity can completely fail the visualization[19]. KLD values for runs with varying perplexity cannot be directly compared since the KLD value is related to perplexity; however, KLD records over time do not show that increased perplexity results in visibly improved data visualization (Fig. 3c) or faster resolution of clusters (Fig. 3d). However, while changing $\alpha$ does not affect t-SNE computation time, perplexity is linearly related to the time required to create the embedding in both implementations of BH-tSNE that we tested. Although we and others have found some benefits of perplexity increases to map quality in otherwise suboptimal t-SNE runs, optimizing the EE

step as described above and further in this work does not leave much space for improvement with perplexity tuning.

**T-SNE learning step size must scale to dataset size**. The step size in t-SNE gradient descent is updated at each iteration per Jacobs adaptive learning rate scheme[22]. This method increases the learning rate in directions in which the gradient is stable. A conservative initial value of 200 is hard-coded into most platforms. We hypothesized that larger datasets may stay longer on KLD plateau due to the number of iterations it takes to build up a sufficient learning rate step size. To evaluate this possibility, we titrated the step size $\eta$ while observing the KLD with fixed EE = 1000 iterations in the mass41parameter dataset. In agreement with our hypothesis, $\eta = 25$ and $\eta = 50$ runs failed to resolve from KLD plateau within 1000 iterations of EE (Fig. 4a) and $\eta = 200$ finished the plateau in ~700 iterations as previously shown. With further increases in $\eta$, we found that not only were progressively fewer iterations required to complete the plateau, but also that the final KLD of the maps scored at lower values. KLD is directly related to the quality of visualization since it reflects the faithfulness of representation of high-dimensional data in t-SNE space; therefore, lower KLD values indicate superior visualization quality.

We continued to see improvement in plateau duration and KLD values with higher $\eta$ values up to $\eta \sim 64{,}000$, a value that is drastically far from the default $\eta = 200$ setting (in most platforms, $\eta$ is restricted to ranges below 3000) (Fig. 4b). At $\eta \sim 256{,}000$ we observed irregular peaks in the KLD graph indicating that the prescribed step size rendered gradient descent ineffective. However, using lower values of $\eta$ we were able to create the map with the lowest KLD values at a fraction of time when limiting the EE step to 200 and even 100 iterations despite the $10^6$ size of the dataset (Fig. 4c). Visual inspection of the embedded maps over the range of $\eta$ values agrees with the KLD values (Fig. 4d).

In a recent publication, Linderman and Steinerberger[23] prove that in general t-SNE embedding will not converge if a product of EE factor $\alpha$ and of fixed learning rate step size $\eta$ is larger than the number of datapoints $n$ (i.e., if $\alpha\eta > n$). Since we employ an adaptive learning rate, our selection of initial $\eta$ value is more forgiving; however, we found the most efficient settings of $\eta$ to be close to $\eta = n/12$ for computations where $\alpha = 12$. Therefore, we

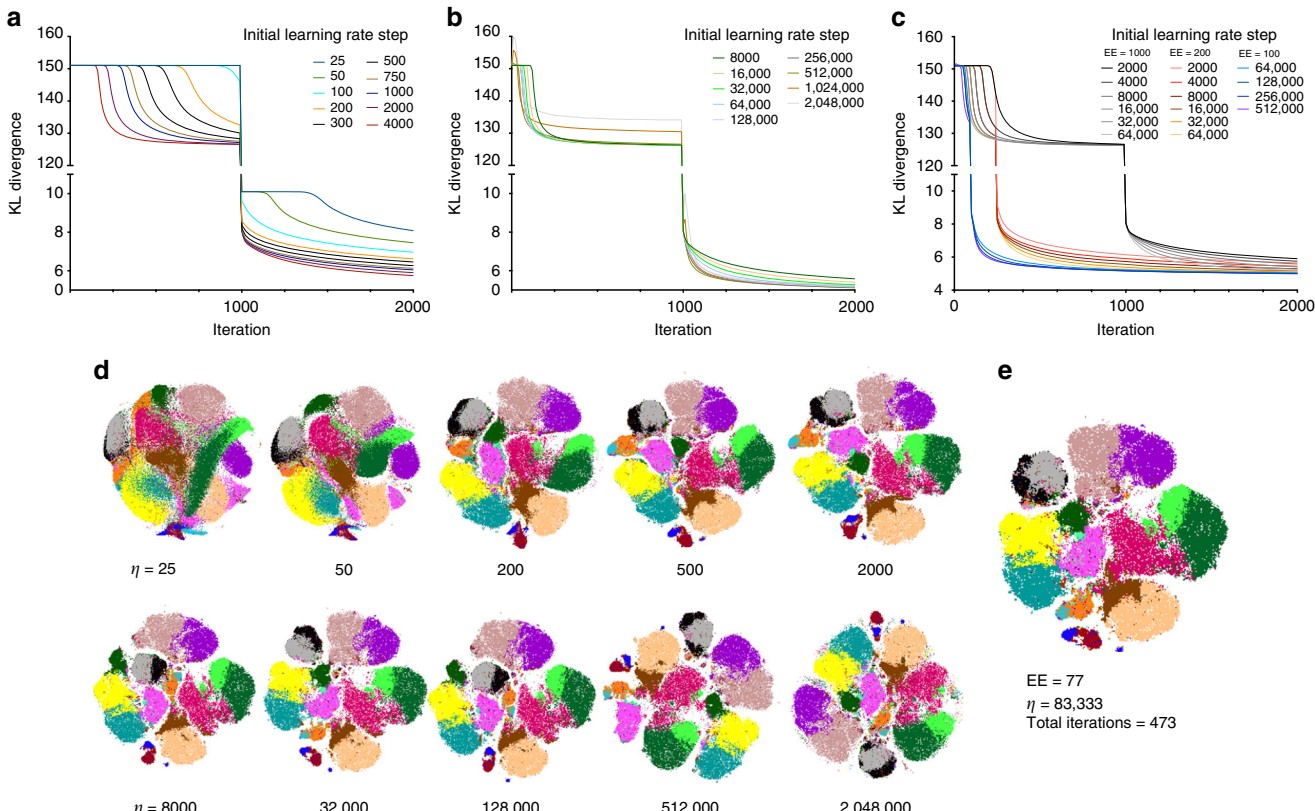

**Fig. 4** Learning step size optimization for t-SNE visualization of large datasets. **a**–**c** KLD change over iterations for embeddings with varying values of initial learning rate step size, color coded as indicated. **a** EE = 1000 iterations, learning rate step = 25–4000; **b** EE = 1000 iterations, learning rate step = 8000–2,048,000; **c** EE = 100–1000 iterations, learning rate step = 2000–512,000. **d** Representative t-SNE plots of embeddings graphed on **a**. **e** t-SNE plot of an optimized embedding. All color overlays on t-SNE plots correspond to cell type classes labeled as in Figs. 1, 2. Representative examples of multiple runs with varying seed values are shown

**Table 2 Summary of t-SNE optimizations proposed in opt-SNE workflow**

| Parameter | opt-SNE setup | Suggested use with cytometry data |
|---|---|---|
| Gradient descent learning rate | Adaptive learning rate with initial value $\eta = n/\alpha$, where $n$ is the number of datapoints and $\alpha$ is the early exaggeration factor | Automated per dataset |
| Early exaggeration factor | Standard t-SNE setup considerations apply | 4–12 |
| Perplexity | Standard t-SNE setup considerations apply | 30–50 |
| Early exaggeration termination | KLD value (cost function) is monitored in real time, and early exaggeration is removed at maxKLDRC | Automated per dataset |
| t-SNE termination | KLD value (cost function) is monitored in real time and the embedding is finalized when $(KLD_{N-1} - KLD_N) < KLD_N/X$ | $X = 5000$ |

propose to initiate the gradient descent with $\eta = n/\alpha$ to create t-SNE visualization (Fig. 4e).

**opt-SNE creates fast high quality embedding of cytometry data**. The standard C++ BH-tSNE implementation that we used to optimize t-SNE parameters only utilizes a single processor core and requires considerable computation time. We thus adopted a recently developed multicore modification of Barnes-Hut t-SNE[24] to implement our proposed optimization techniques into a single workflow labeled 'opt-SNE' that includes KLD plateau monitoring and dataset size-specific learning rate step scaling (Table 2). To compare opt-SNE to standard t-SNE and individual t-SNE parameter tuning results, we generated embeddings that were

automatically finalized when $(KLD_{N-1} - KLD_N) < KLD_N/5000$ (i.e., when each new iteration only improved KLD by less than 0.05%). When we moved from original C++ BH-tSNE to multi-core BH-tSNE[24], we observed a 2–3× boost in computation speed with no penalty in embedding quality, endpoint KLD values, or total number of iterations to reach endpoint KLD (Suppl. Fig. 4A). When compared with standard a t-SNE run in multicore enviroment, opt-SNE visualizations of both flow and mass cytometry data exhibited an additional 2× or greater speedup due to the smaller number of iterations needed to complete the data embedding (Fig. 5b, Suppl. Fig. 4B). This improvement in computation time scaled with dataset size, and almost no variation was observed when runs initiated from various random seeds were compared (Suppl. Fig. 4B-E). As discussed above, opt-SNE

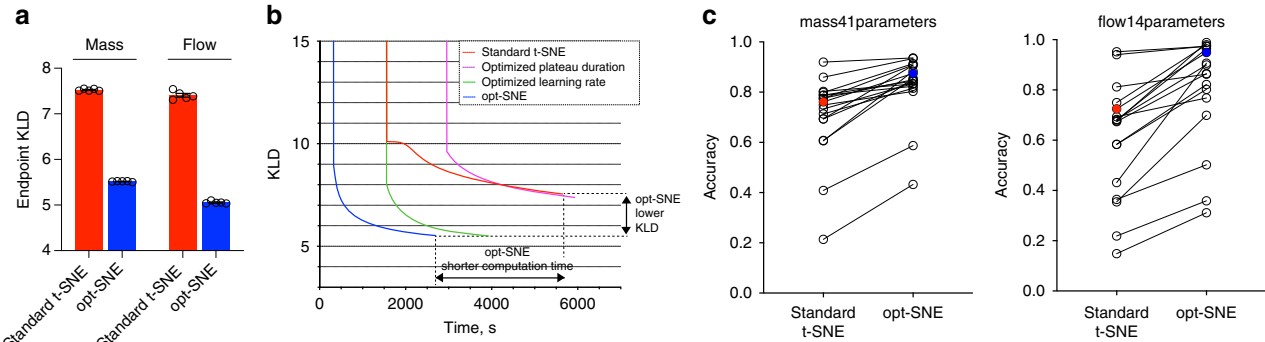

**Fig. 5** Evaluation of opt-SNE embeddings. **a** Endpoint KLD values for standard t-SNE (initial learning rate step = 200, EE stop = 250 iterations) and opt-SNE (initial learning rate = n/α, EE stop at maxKLDRC iteration). $N = 5$ seeds used for random initialization; error bars denote SEM. **b** Post-EE graph of KLD minimization over physical time for standard t-SNE, adjusted parameter (as indicated) t-SNE and opt-SNE (representative examples of mass cytometry data embeddings are shown). **c** 1NN accuracy scores for standard t-SNE and opt-SNE embeddings of of mass cytometry (left) and flow cytometry (right) data per assigned class values (cell subsets, open circles; overal scores, filled circles). Representative examples of multiple runs initiated with varying seed values are shown

maps demonstrated superior quality compared to standard t-SNE embeddings when visually inspected. We also assessed the quality and accuracy of the embeddings quantitatively by comparing their endpoint KLD values and 1NN accuracy scores. opt-SNE embeddings achieved values of endpoint KLD that were significantly lower than in standard t-SNE (Fig. 5a, b) and such values were not reached by the standard t-SNE calculation even after 20,000 iterations, indicating better retention of multi-dimensional data structure. When compared on 1NN classifier accuracy, opt-SNE embeddings achieved better scores than embeddings produced with standard t-SNE methodology (Fig. 5c). Also, by the same metric, the $KLD_N/5000$ endpoint criterion has proven to mark sufficiently long opt-SNE runs since no improvement of accuracy was observed when the embedding calculation continued beyond that point (Suppl. Fig. 4E). For further algorithm tuning at the discretion of the user, we incorporated automated opt-SNE termination as a tunable parameter that sets a fraction of KLD minimization per iteration.

**opt-SNE allows visualization of mega-scale single cell data**. To test the performance of opt-SNE on even larger datasets, we used a $20.1 \times 10^6$ event fluorescent cytometry dataset concatenated from two independent batches of peripheral blood mononuclear cells (PBMC) samples ($N = 27$) stained with a variation of the OMIP-037 fluorescent cytometry panel[25] that allows detailed assessment of multiple immune subsets including naïve and memory CD4+ and CD8+ T cells, NK cells, and γδ T cells (Fig. 6a, b). The embedding completed in 770 iterations with only 73 iterations required to pass the EE step (Suppl. Fig. 5A) and resulted in clear separation of cell clusters as evaluated by cell type annotation (Fig. 6c). The majority of clusters appear to be populated by cells from all subjects with the exceptions of several populations that contained sample-unique debris features (Fig. 6b, dashed arrows), confirming an absence of batch effects. A detailed breakdown of identified populations is presented in Fig. 6c that shows subsets of CD4+ and CD8+ T cells, NK cells, γδ T cells, B cells, and monocytes. Importantly, B cell and monocyte lineage markers were detected together with a viability dye in a dump channel in this panel and therefore could not be gated accurately via traditional biaxial plot analysis. However, opt-SNE helped to visualize these populations as clear cell groupings in the embedding analogously to their presence in the original high-dimensional space defined by the combination of

several surface labels and light scatter characteristics, and the resulting opt-SNE islands were minimally mixed with dead cells. Use of the standard t-SNE algorithm on this dataset completely failed to reveal its structure, even with several thousands of iterations (Fig. 6d).

To test the suitability of opt-SNE for applications beyond flow and mass cytometry, we analyzed a $1.3 \times 10^6$ cell single cell RNA-seq dataset of mouse embryonic brain cells published by 10x Genomics. We used pre-calculated PCA projections included in the dataset to generate opt-SNE maps that we compared with 10x Genomics standard t-SNE embedding (Fig. 6e). 10x Genomics used EE = 1000/total 4000 iterations of standard t-SNE while we used opt-SNE settings with $\eta = 97,959$, EE = 66/total 885 iterations (Suppl. Fig. 5B). Non-immune single cell transcriptomics data are more difficult to interpret with ground-truth classes since much fewer scRNA-seq markers can be easily interpreted for population identification. Therefore, we utilized both single gene classification and classification through the Louvain clustering algorithm using the Scanpy Python package[26] to annotate the data. While opt-SNE and t-SNE both capture the macro-structure illustrated by gene overlays, several Louvain clusters (3, 18, 19, 25, 28) appear partially or completely overlapped by other clusters in standard t-SNE but not in the opt-SNE embedding. Therefore, opt-SNE allowed equivalent or superior resolution of single cell transcriptomics data as with standard t-SNE but with ~5× smaller iteration time (885 iterations of opt-SNE vs. 4000 iterations of standard t-SNE).

HSNE (hierarchical SNE) is a t-SNE adaptation that was recently reported to facilitate analysis of large datasets by constructing a hierarchy of embeddings that can be explored from the overview of landmark populations up to a single cell level or resolution[7,17]. To directly compare HSNE and opt-SNE, we applied opt-SNE to the 5.2 million point dataset that was reported in HSNE analysis of mass cytometry data[7], and compared opt-SNE visualization to the full resolution level of HSNE embedding (Fig. 6g, Suppl. Fig. 6). In the CD4+ subset, opt-SNE visualization revealed two groups of CD4+CD28− cells likely representing terminally differentiated memory CD4+ T cells[27] with different levels of CCR7 expression that may reflect distinct differentiation states of the two populations. While HSNE allowed identification of CD4+CD28−CCR7− cells, it was unable to visualize the CD4+CD28−CCR7+ cells as a single cluster and projected these cells sparsely in the CD4+ island (Fig. 6i, left). Also, both algorithms projected heterogenous

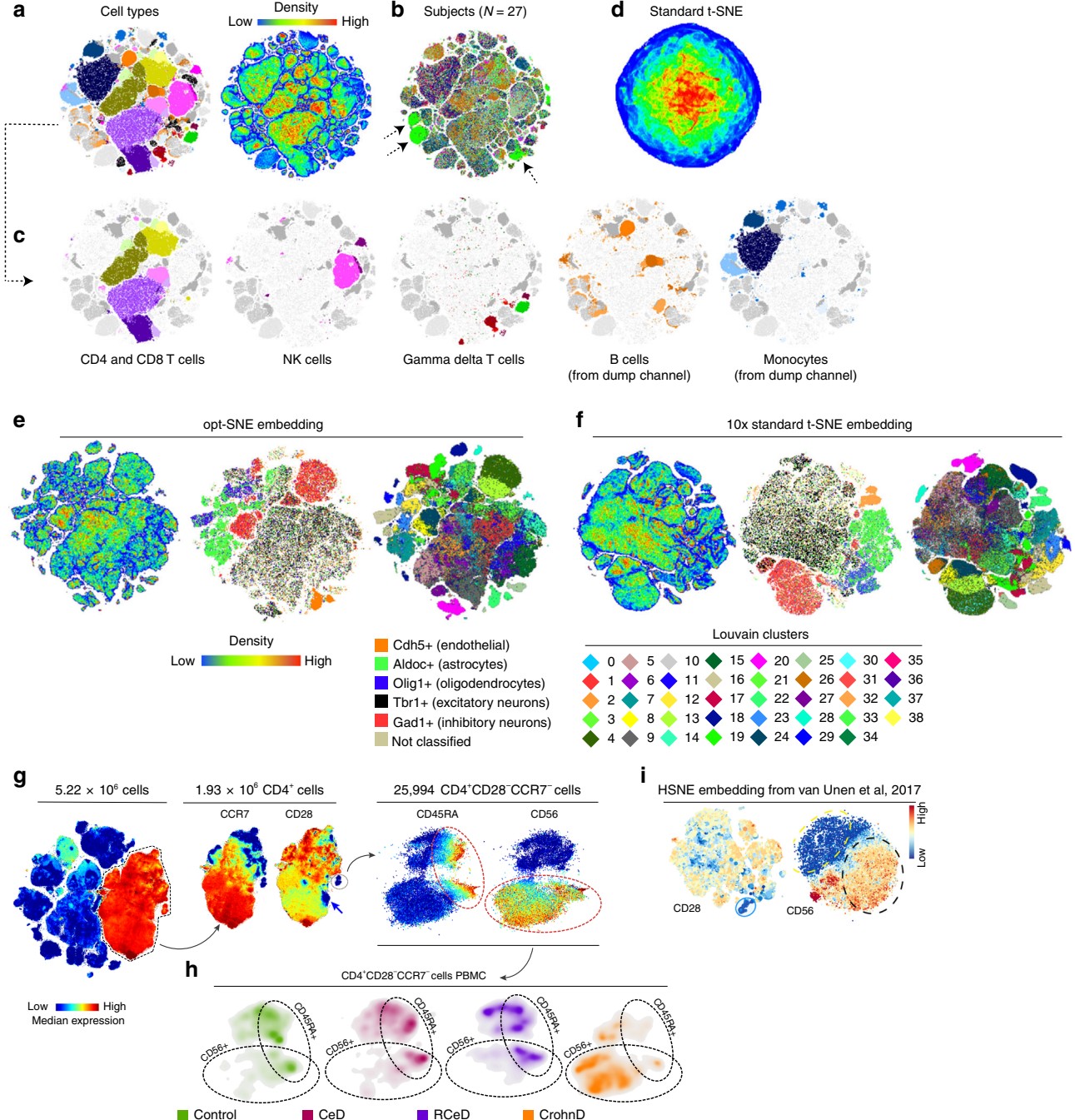

**Fig. 6** opt-SNE allows high-quality visualization of large cytometry and transcriptomics datasets. **a–d** 20 million datapoints from fluorescent cytometry dataset concatenated from 27 subjects vizualized in 2D space. **a, c** Cell type classes and density overlaid on 2D opt-SNE embedding. **b** Subject identifier overlaid on 2D opt-SNE embedding. Dashed arrows indicate clusters represented by datapoints from a single subject. **d** Standard t-SNE visualization (4000 iterations). **e, f** 10x Genomics mouse brain scRNA-seq dataset (1.3 million datapoints) visualized in 2D space with opt-SNE (**e**) or standard t-SNE (**f**). From left to right: density features, single gene classes, and Louvain clusters (0–38) overlays. **g** 5.22 million datapoints from mass cytometry dataset used in van Unen et al (2017) visualized in 2D space with opt-SNE. From left to right: CD4 expression overlaid on opt-SNE embedding; CCR7 and CD28 expression overlaid on CD4$^+$ opt-SNE cluster; CD45RA and CD56 expression intensity overlaid on CD4$^+$CD28$^-$CCR7$^-$ cluster. **h** CD4$^+$CD28$^-$CCR7$^-$ cells from control, celiac disease (CeD), refractory celiac disease (RCeD) and Crohn's disease (CrohnD) subjects presented on density plots. Dashed encirclements indicate CD45RA$^+$ and CD56$^+$ areas of the cluster as defined in **g**. **i** Hierarchical t-SNE (HSNE) embedding of the CD4 (left) and CD4$^+$CD28$^-$ cluster (right) reproduced from van Unen et al.[7] (licensed under a Creative Commons Attribution 4.0; http://creativecommons.org/licenses/by/4.0/). Color indicates marker expression intensity

expression of CD56 in CD4$^+$CD28$^-$CCR7$^-$ cells, but HSNE embedding did not resolve separate CD56$^+$ and CD56$^-$ clusters within CD4$^+$CD28$^-$CCR7$^-$ cells, loosely mapping them to the poles of the single round cluster (Fig. 6i, right). However, opt-SNE embedding visualized both the CD56+ and CD56− clusters and the disparate CD45RA expression within each cluster, revealing distinct phenotypes for the control and diseased subject groups (Fig. 6h). Also, the opt-SNE maps revealed that the CD56$^+$CD45RA$^-$ cells in the cluster originate from several subjects with Crohn's disease (Fig. 6h). In summary, these results confirm that opt-SNE embedding provides superior visualization quality for complex cytometry data.

## Discussion

Visual exploration of data drives hypothesis formation and/or serendipitous discoveries; therefore, t-SNE is an extremely valuable tool for data comprehension. It is often used to facilitate data perception when hypothesis generation is automated by robust computational methods[11,28]. Comparison of t-SNE embeddings from multiple experimental conditions, timepoints, or subjects is invaluable to visualize sample-to-sample differences including disease hallmarks and longitudinal observations[29]. t-SNE is also valuable for quality assessment of data, when abnormal clustering can be traced back to sample preparation, data acquisition, and preprocessing artifacts[30]. Therefore, batch embedding of multiple experimental points is essential for sample comparison and can only be enabled when t-SNE accommodates large datasets.

t-SNE was first introduced in cytometry research as a tool to visualize CyTOF data, since fluorescence-based high-parameter datasets were less common at that time. With recent advances in instrumentation and reagent availability, flow cytometry datasets with >20 parameters are quickly becoming prevalent and even standard in the field[31–33], yet the proper data assessment tools are lacking for general use. DNA-barcoded antibodies have been recently utilized to allow simultaneous protein-epitope and transcriptome measurements in single cells[34] thus expanding the repertoire of traditional cytometry methods that could employ t-SNE as a staple method of data visualization and presentation.

One approach to large dataset t-SNE embedding is to model visualization with a subset of datapoints curated[7,17] or randomly selected (as implemented in cytometry data analysis platforms such as FCS Express and Gemstone) from the nearest neighbor graph. Such techniques may fail to project extremely rare datapoints, as demonstrated by the CD4$^+$CD28$^-$CCR7$^+$cells that were identified via use of opt-SNE, but not HSNE, in the van Unen et al dataset[7] (Fig. 6g). This T cell subset was significantly less abundant in subjects with severe inflammatory conditions including Crohn's disease as compared to controls (Suppl. Fig. 6). However, the cells expressing the CD4$^+$CD28$^-$CCR7$^-$ phenotype, while also low in frequency in most PBMC samples (<1% of all CD4$^+$ cells), was likely successfully clustered by HSNE because two of the 11 Crohn disease subjects in the dataset showed an unusually high frequency of this population (40.7% and 31.0% of all CD4$^+$ cells), allowing it to be well represented in the kNN graph of the full dataset.

Several attempts to successfully apply t-SNE-like methods to massive datasets have been recently reported including aforenoted HSNE[7,17,35], LargeVis[8], and net-SNE[36]. When applied to large datasets, these methods often require and/or benefit from considerable computational resources; for instance, the LargeVis study was performed on a 512 GB RAM, 32 core station. However, it is ideal for such data analysis to be performed successfully with the computational resources found in most laboratories. We benchmarked our opt-SNE modification of multicore t-SNE C++ implementation on a quad-core, 16GB RAM workstation

although even further improvement in computation speed would occur if more cores were available. FIt-SNE, a recently published alternative to Barnes-Hut t-SNE that uses fast Fourier transform for much faster computation of repulsive forces approximation[18], renders opt-SNE even more feasible on personal computers. Notably, we have not observed differences in quality between embeddings generated with fast Fourier transform vs. Barnes-Hut approximations when opt-SNE parameter adjustments were used to compute the visualization (Suppl. Fig. 7). Therefore, we expect opt-SNE to be applicable for existing or future adaptations of t-SNE even if alternative methods of computation are utilized[37,38] provided that they retain the core principles of t-SNE embedding. A promising approach that may be integrated with opt-SNE is the smart EE adjustment implemented in A-tSNE (approximated t-SNE)[35] algorithm where EE is removed gradually and on a per-point basis. Cytosplore, a novel software platform that includes HSNE and A-tSNE, allows the analyst to interactively initiate the local refinement of the map, resulting in a significant improvement in computation time[7].

Often compared to t-SNE, Uniform Manifold Approximation and Projection (UMAP) is a recently developed novel technique for dimension reduction that can be successfully applied to cytometry datasets[9]. As has been the case for other groups[9,21], our comparison of t-SNE and opt-SNE to UMAP generated mixed results. Specifically, we found that UMAP parameter selection had dramatic effects on UMAP computation performance and embedding quality assessed by both human perception and 1NN classifier accuracy (Supp. Fig. 8A-E). When benchmarked for resource use in the Omiq cloud, UMAP consumed up to 128 GB of RAM when embedding 5.2 million cell van Unen dataset, while opt-SNE and FIt-SNE peaked at 16 GB RAM consumption; the UMAP embedding took a comparable amount of time to complete as opt-SNE. With careful parameter selection, UMAP accuracy was approaching, but not surpassing, opt-SNE accuracy (Supp. Fig. 8A, B).

Similar to other types of biological data, the structure of cytometry data is difficult to project due to its mixed nature, often comprised of cluster-like, manifold-like and/or hierarchical components[30,39]. In this paper we propose multiple techniques that are essential for superior quality t-SNE data projection and are all germane to the fine-tuning of the early exaggeration phase of t-SNE embedding. EE facilitates cluster formation on a 2D plane[1,40] and serves as a necessary compromise that allows clusters to escape the crowding effect. We designed an efficient measure to ensure that the cluster-like global structure of the data is fully revealed during the EE phase by monitoring the KLD output of the embedding in real time. As indicated by the lower KLD values of opt-SNE embedding where EE was limited to fewer iterations (Fig. 4e), prolonged amplification of the attractive forces that drive tight cluster formation in EE may be detrimental for the manifold-like local data structure represented by signal distributions that are continuous with background signal. These 'continuum expression' molecules include proteins used to define classic immune cell subsets as well as markers indicating disease phenotypes, cell activation, and/or exhaustion state[41]. Conversely, the non-exaggerated phase of t-SNE allows local data structures to be revealed[1]. Therefore, cytometry data analysis would be missing valuable information if we reduced t-SNE applicability to finding only well separated clusters, especially since other techniques would perform this task better and faster. However, some workflows call for t-SNE pre-processing to facilitate extraction of cluster features from multidimensional data[5,42]. In those cases, it may be helpful to adapt the opt-SNE toolkit to terminate the embedding calculation immediately at the EE stop iteration and re-assess the high-dimensional structure within each cluster.

Alternatively, a 'late exaggeration' approach[21,38] can be cautiously applied to create very tight clusters, although in our experience this approach was only marginally beneficial for global structure representation while detrimental for local structure (Suppl. Fig. 9). Interestingly, UMAP embeddings also presented a challenge to perceive both cluster data structure and continuum marker expression data structure within the same embedding, as increasing the number of neighbors and decreasing the minimum distance parameters seemed to benefit the former but penalize the latter task (Suppl. Fig. 8C-E and[21]). These results suggest that UMAP is highly promising but not yet readily available to replace t-SNE as the state-of-the-art method for cytometry data visualization.

It is advisable to note that certain data structures, such as cluster hierarchy, cannot be revealed with t-SNE[43,44]. t-SNE accessibility in cytometry analysis software lead to its not infrequent misuse with cytometry data, evident when the cluster-like structure is not prominent in the map. Therefore, the features identified from t-SNE embedding in its current form should be verified with alternative methods when possible for confirmational purposes. Im et al also suggest that if a continuous manifold structure exists in the data, large perplexity values may cause artificial breaks (overclustering)[44]. The perplexity values commonly used in cytometry analysis are on the lower end of the suggested range for efficient clustering, as it is often advised to scale the number of nearest neighbors to the average cluster size[45]; however, if computationally feasible, higher perplexity values might facilitate feature preservation for markers whose expression is not bimodally distributed. To preserve global geometry, Kobak et al[21] propose to perform the embedding in two steps and use the kNN-based extrapolation of a high-perplexity t-SNE of a subset of the large dataset as initialization to run the t-SNE of the whole dataset.

In summary, opt-SNE is a powerful optimization toolkit that subverts major limitations of t-SNE for use with cytometric datasets and thus enables novel data-driven findings in single cell research.

## Methods

**Datasets**. All datasets used in the study are summarized in Table 1.

**Data pre-processing**. Singlet events from several data recordings were digitally concatenated and a randomly subsampled file of 1,000,000 mass[46] or flow cytometry[47] events was created and used for analyses of the mass41parameter and flow18parameter datasets. All observations from 27 recordings of flow cytometry data were concatenated to generate the flow20M dataset. All observations from 104 recordings of mass cytometry data[7] were concatenated to generate the van Unen et al dataset.

All flow cytometry data were compensated with acquisition-defined compensation matrices. Prior to t-SNE analysis, all cytometry data were transformed using asinh (with cofactor 5 for mass cytometry data and with cofactor 150 for flow14parameter data) or biexponential (flow20M) transformation. Light scatter parameters were log-transformed.

**The standard t-SNE configuration for cytometry applications**. t-SNE computes low-dimensional coordinates of high-dimensional data resulting in similar and dissimilar data points in the raw data space placed proximally and at a distance, respectively, in the dimensionally reduced map[2]. This map placement is achieved via t-SNE modeling the probabilities as a Gaussian distribution around each data point in the high-dimensional space and modeling the target distribution of pairwise similarities in the lower-dimensional space using Cauchy distribution (Student t-distribution with 1 degree of freedom). Then, the Kullback-Leibler Divergence (KLD) between the distributions is iteratively minimized via gradient descent. The gradient computation is essentially an N-body simulation problem with attractive forces (approximated to nearest neighbors using vantage-point trees) pulling similar points together and repulsive forces (approximated at each iteration using the Barnes-Hut algorithm) pushing dissimilar points apart.

An important part of t-SNE gradient descent computation is the early exaggeration (EE) that was proposed by van der Maaten and Hinton[1] to battle the so called overcrowding artifact of embedding. With EE, all probabilities modeling distances in high-dimensional space are multiplied by a factor (early exaggeration factor, EEF, or $\alpha$) for the duration of the EE phase (typically 250, or 25% of the total number of iterations). EE coerces data to form tight and widely separated clusters in the map and is considered to enable the map to find a better global structure.

Multiple software platforms incorporate the BH-tSNE algorithm specifically for analysis of cytometry data, including commonly used cytometry analysis desktop packages (FlowJo, FCS Express), and the cloud-based analysis platform Cytobank. Also, implementations of t-SNE are available as open-source packages in popular programming languages such as R (rtsne) and Python (sci-kit learn). Most of implementations wrap or re-write original C++ code of Barnes-Hut t-SNE[2] and produce comparable analysis results upon direct comparison. Here, we customized the standard t-SNE C++ code to implement the parameter adjustments described in this work and published this customization as an open source solution to enable the research community to use this optimized t-SNE algorithm. Also, the equivalent adjustments of t-SNE have been made available as a cloud application in Omiq and integrated into FlowJo and SeqGeq programs.

**Data analysis**. A desktop C++ Barnes-Hut implementation of t-SNE for Mac OS was used for t-SNE analyses[2] with original implementation adapted to allow user input for the early exaggeration stop iteration, perplexity, total number of iterations, early exaggeration factor value, and learning rate value. A multicore modification of Barnes-Hut t-SNE[24] was used as a source for building the opt-SNE package with user-accessible parameters similar to listed above. The Kullback-Leibler Divergence (KLD) value and t-SNE coordinates were reported during each iteration or as frequently as requested. All datasets were embedded in 2D space. Visually comparable t-SNE maps were generated with the same random seed values used when permutations for specific parameters were tested and compared; all experiments were repeated with several values of random seed. Benchmarking data were generated on an iMac personal station with Intel Core i7 quad-core i7 processor and 16 GB of RAM. For cross-validation, we utilized cloud-based Cytobank[48], cloud-based Omiq, FlowJo V10.3–10.5 and FlowJo V9.9.6. R flowcore package, Cytobank and FlowJo were used to generate FCS files and graphical outputs from tabular data. Omiq platform was used for FlowSOM[11], FIt-SNE (Fast Fourier Transform-accelerated Interpolation-based t-SNE[18]) and UMAP (Uniform Manifold Approximation and Projection)[9] analyses. Also, the FIt-SNE plugin for FlowJo and FIt-SNE prototype in FlowJo were used to ensure FIt-SNE compatibility with opt-SNE. Logs of t-SNE runs were batch-processed with VBA scripts and analyzed with GraphPad Prism. Expert-guided (manual) analysis of cell populations was performed in FlowJo 10.3–10.5 as described previously for specific datasets[25,46] (Suppl. Figs. 10–11) and used for map annotations as cluster classes.

We used SeqGeq 1.3 for scRNAseq analysis and leveraged PCA projections provided in the 10x Genomics dataset to calculate the t-SNE embedding and annotated it using marker genes for major cell types. Louvain cluster classification was adopted from the SCANPY data analysis study[26].

The quality of embedding was assessed by a 1-nearest neighbor (1NN) classifier based on previous reports on t-SNE accuracy evaluation[1,2,8]. The classifier was written in R (version 3.5.0). For each invocation, the classifier was provided a complete embedding and an accompanying array of assigned class values for each observation in the embedding. Ten thousand cells were sampled from the embedding as a training set using uniform pseudorandom sampling with the "sample" function from base R. Next, a set of 50,000 cells disjoint to the training set was sampled from the embedding as the test set. Each cell in the test set was then mapped to its single nearest neighbor within the test set and assigned the class of that neighbor using the "knn" function of the "FNN" R package (version 1.1.3, available on CRAN). The number of correct class assignments in the test set was calculated on a total basis and also on a per-class basis using the supplied class values. This process was repeated five times for the provided embedding using different seeds for the sampling step. Replacement was allowed for pseudorandom sampling between the five internal repeat runs. The results were then presented as the average of these five repetitions. The seeds used for the five repetitions were kept consistent such that classification results between different embeddings of the same dataset tested the same cells.

**Reporting summary**. Further information on research design is available in the Nature Research Reporting Summary linked to this article.

## Code availability

Open source multicore t-SNE C++ implementation and usage instructions are available at https://github.com/omiq-ai/Multicore-opt-SNE. Cloud version of opt-SNE is available at http://www.omiq.ai/opt-SNE. opt-SNE is natively supported in the FlowJo data analysis software version 10.6 and later, and the SeqGeq version 1.5 or later, both downloadable from http://www.flowjo.com. The code of modified van der Maaten BH-tSNE C++ implementation with user-accessible parameters is available upon request.

## Data availability

Flow18parameters and mass41parameters datasets are available via FigShare at https://doi.org/10.6084/m9.figshare.9927986.v1. Van Unen et al. dataset[7] is available at http://

flowrepository.org/id/FR-FCM-ZYRM. 10XGenomics 1.3M scRNA-seq dataset is available at https://support.10xgenomics.com/single-cell-gene-expression/datasets.

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

## Acknowledgements
The authors would like to thank Yvan Saeys, Gary Kazantsev, Jonathan Irish, Dmitry Kobak, Allison Irvine and Katherine Drake for helpful discussions, and Geoff Kraker for technical assistance. We thank Sean Bendall from Stanford University and Vincent van Unen from Leiden University Medical Center for sharing their data, and Brian Tilton and Riley Pihl from BUSM Flow Cytometry Core Facility for assistance with data collection at BUSM. Anna C. Belkina is an ISAC (International Society for Advancement of Cytometry) SRL Emerging Leader and thanks the ISAC organization and members for continuous support and encouragement. Josef Spidlen is an ISAC Marylou Ingram Scholar.

## Author contributions
A.C.B. has conceived the study. A.C.B. and C.O.C. developed the opt-SNE method. C.O. C. developed implementation in C++; and R.H. and J.S. developed implementation in FlowJo and SeqGeq. A.C.B. has analyzed flow and mass cytometry datasets. R.H. and J.S. have analyzed the scRNA-seq dataset. R.A. and J.S.C. provided conceptual input. A.C.B. wrote the paper. All authors discussed the results and commented on the paper.

## Competing interests
C.O.C. is a founder of Omiq, Inc. R.H. and J.S. are employees of Beckton Dickinson (BD); FlowJo is a subsidiary of BD. The remaining authors declare no competing interests.
