## [Peer Review File · Nature Communications]

Reviewers' comments:

Reviewer #1 (Remarks to the Author):

The authors investigated the various parameters of the t-distributed Stochastic Neighbor Embedding (t-SNE) algorithm in the context of mass cytometry, flow cytometry, and scRNAseq data sets with millions of points. They identified that reaching the maximum of the Kullback-Liebler Divergence (KLD) measure during the early exaggeration phase of the algorithm's gradient descent indicates resolution of the global structure of the data. Therefore, the early exaggeration phase can be stopped once the maximum is reached. Following this observation, the authors tested the alpha value, number of iterations, and step size parameters, and offer recommendations for each. They implemented these heuristics in a software package named opt-SNE and showed that they lead to high-quality t-SNE maps in two large data sets.

t-SNE is one of the most popular methods for visualizing single-cell data. To the best of my knowledge this is the only study into the effect of parameterization on the results of the method in biological contexts. The authors reach several unexpected conclusions that are bound to influence future use of t-SNE.

My comments of the manuscript follow.

Introduction

- The introduction discusses PCA and t-SNE but makes no mention of other dimensionality reduction methods. I think that the authors should include a paragraph that discusses UMAP, LargeVis, HSNE, and potentially force-based layouts of graph.
- Although this is not a peer-reviewed article, <https://distill.pub/2016/misread-tsne/> has been influential of people's perception of t-SNE and might be worth including here.

Methods

- For the asinh transformation, which cofactor did the authors use? (If at all.)
- In line 118, the "KLD" acronym is used before being defined.
- "The standard t-SNE configuration for cytometry applications", in my opinion this section can arrive earlier in the methods, as it frames them nicely.
- Please include exact specification of the computers used in this analysis. They are briefly mentioned in the discussion but a more precise spec is required.

Results

- Most notably, the authors include a NN-based accuracy score in the methods section but then ignore it for most of the text. I greatly appreciate the many visuals included but the quantification is missing. Please apply the NN-based accuracy score to Figure 1 and to Figure 6 (for Figure 1 it is briefly mentioned in the text but is difficult to find). Notably, for Figure 6, the metric can be used to compare opt-SNE and HSNE.
- In the first section, "The standard t-SNE configuration fails to visualize large datasets", the authors should include at least one other method as comparison. I recommend UMAP as it is gaining popularity in the field.
- All of the figures are too small. For example, in Figure 1, I appreciate the better separation of the

3,000 iteration plot, but in current image size this corresponds to five pixels. I think that the figures can be restructured into full-page figures (instead of taking only half a page), and that some panels can be reduced (for example, show outputs over less parameter values) and moved to the supplement.

- For figure 4, I think that there is a mismatch between figure order and text order: 4B is referenced in the text before 4A. Additionally, there is a reference in the text to 4E, which does not exist.

- Could the authors please include a table that summarizes their recommendations regarding each of the parameters? The values are spread across the text and it would be beneficial to have a summary either here or in the discussion.

Discussion

- One of the greatest selling points of UMAP is that it provides a better view of the global structure of the data compared to t-SNE. Do the authors think that opt-SNE bridges that gap? Either way, I think that this point deserves mentioning here.

Signed,
El-ad David Amir

Reviewer #2 (Remarks to the Author):

The article by Belkina et al. describes an algorithm for identification of "optimal" parameters for t-SNE. The article was difficult for us to read as it is presented as a step by step "story" of anecdotal examples and "visual inspections". Unfortunately, despite bold claims, the article does not provide sufficient objective information to support said claims. We have listed some of our major comments below:

1) Starting with the title and name "Opt-tSNE": The claim of a parameter value being optimal is a bold claim that must be taken seriously and supported by evidence. Calling something optimal when evidence at best suggests that it may be an "improvement" would come across as non-scientific, if not misleading. Another example of this is line 264 "... since they provided optimal balance of map quality versus computation time." The theme of making strong statements without clear scientific evidence repeats throughout the manuscript. We have not provided a comprehensive list in this review.

2) Re. the claim of "dramatic improvement of computation time": There is no comparative study of computation times and the discussion even states that "we have not explicitly focused on efficiency in this work since we benchmarked the algorithm against itself with no specific emphasis on shortening computation time". To validate the claim on computation time improvements, a comprehensive study across several datasets and settings would be required. Especially since on first sight the proposed real-time KLD monitoring approach can potentially result in infinite runtimes (unless there are other control parameters that are not discussed - in which case analysis of those parameters would also be required). This is even further complicated since the results of all these analysis are different and given the unsupervised nature of the data, by definition, there is no way to determine which solution is better and by how much, therefore any runtime comparison would be compromised from the beginning.

3) Re. the claim of "optimal data resolution": There is no quantitative measure of "optimal data resolution" as well as "precise data interpretation". The results claimed are based on visual inspection only and reportedly use "representative examples of multiple runs with varying seed values". These are simply the authors opinions not scientific evidence backed by empirical data

and statistical analysis.

4) Initialization of machine learning algorithms has been subject to research for decades. It is well known that improved initialization can decrease convergence time. Similarly, a convergence criteria to pause the training process is an idea that has long existed in the field, numerous solutions have been developed, and even some existing tSNE packages already implement. Without objective and quantitative comparison of several state-of-the-art initialization and convergence detection techniques, the authors are simply reporting software development improvements not scientific studies.

Response to the Referees

Reviewer #1:

The authors investigated the various parameters of the t-distributed Stochastic Neighbor Embedding (t-SNE) algorithm in the context of mass cytometry, flow cytometry, and scRNAseq data sets with millions of points. They identified that reaching the maximum of the Kullback-Liebler Divergence (KLD) measure during the early exaggeration phase of the algorithm's gradient descent indicates resolution of the global structure of the data. Therefore, the early exaggeration phase can be stopped once the maximum is reached. Following this observation, the authors tested the alpha value, number of iterations, and step size parameters, and offer recommendations for each. They implemented these heuristics in a software package named opt-SNE and showed that they lead to high-quality t-SNE maps in two large data sets.

t-SNE is one of the most popular methods for visualizing single-cell data. To the best of my knowledge this is the only study into the effect of parameterization on the results of the method in biological contexts. The authors reach several unexpected conclusions that are bound to influence future use of t-SNE.

This is an accurate summary of our results and we thank the reviewer for his comments highlighting the significance and impact of our findings.

My comments of the manuscript follow.

Introduction

- The introduction discusses PCA and t-SNE but makes no mention of other dimensionality reduction methods. I think that the authors should include a paragraph that discusses UMAP, LargeVis, HSNE, and potentially force-based layouts of graph.

We have added text that mentions alternative dimensionality reduction methods to the introduction (lines 50-52, 78-82) and we have significantly expanded the section of the Discussion that describes other dimensionality reduction methods, including UMAP.

- Although this is not a peer-reviewed article, <https://distill.pub/2016/misread-tsne/> has been influential of people's perception of t-SNE and might be worth including here.

We agree about the importance and relevance of that publication and in fact it was listed as a reference in the manuscript ("Wattenberg, et al., 'How to Use t-SNE Effectively', Distill, 2016.") This reference is retained in the revised manuscript as Ref. 25 (lines 182, 273).

- For the asinh transformation, which cofactor did the authors use? (If at all.)

For the mass cytometry and flow cytometry datasets we used cofactors of 5 and 150, respectively, for the asinh transformations. We have now included this information in the Methods section, lines 110-111).

- In line 118, the "KLD" acronym is used before being defined.
We have now added the definition of KLD with its first use in the text.

- "The standard t-SNE configuration for cytometry applications", in my opinion this section can arrive earlier in the methods, as it frames them nicely.

We have re-ordered the text in the Methods as requested.

- Please include exact specification of the computers used in this analysis. They are briefly mentioned in the discussion but a more precise spec is required.

The specifications of the computer resources used throughout the paper have been added to the Methods section as requested.

Results

- Most notably, the authors include a NN-based accuracy score in the methods section but then ignore it for most of the text. I greatly appreciate the many visuals included but the quantification is missing. Please apply the NN-based accuracy score to Figure 1 and to Figure 6 (for Figure 1 it is briefly mentioned in the text but is difficult to find).

We have revised our algorithm evaluation strategy to now provide more detailed quantifications in the revised manuscript. The 1NN classifier accuracy score has been calculated for standard t-SNE and opt-SNE embeddings of mass and flow cytometry datasets used throughout the paper. These scores are now presented in Fig. 5 and Suppl. Fig. 1, 4, 8. Specifically, detailed Figure 1 scores are now reported in Suppl. Fig. 1 and Fig. 6 scores are reported in Suppl. Fig. 8. These new data, based on similar methodologies used in recent relevant papers, provide quantitative comparison of the quality of embeddings generated with standard t-SNE and opt-SNE parameters.

Notably, for Figure 6, the metric can be used to compare opt-SNE and HSNE.

Unfortunately, the hierarchical nature of HSNE prohibits assessing the detailed embedding with any metric including 1NN classifier score because the full dataset embedding does not exist. In HSNE, the dataset is embedded first at a so-called 'overview' level using 'landmarks' of subsets of the dataset. Further subsets can be chosen manually by the analyst to repeat this process, eventually arriving at an embedding of single-cell resolution. Unlike other dimension reduction algorithms which have unbiased inputs and outputs and thus have an intuitive basis of comparison, HSNE requires subjective exclusion steps and only provides embeddings of portions of a dataset at a time. The original Cytosplore/HSNE manuscript (van Unen et al, 2017) does not include the raw data files or exclusion criteria for the "third level HSNE embedding" that we then cited in our paper, and this fact has restricted us to the use of graphical visualizations generated by van Unen et al analysts for comparisons of HSNE and opt-SNE.

Cytosplore/HSNE in its current form employs standard Barnes-Hut t-SNE settings. We chose it to be compared to opt-SNE in our manuscript since van Unen et al is the only peer-reviewed publication to date that describes dimensionality reduction method applied to such a large (over 5 million datapoints) mass cytometry dataset.

- In the first section, "The standard t-SNE configuration fails to visualize large datasets", the authors should include at least one other method as comparison. I recommend UMAP as it is gaining popularity in the field.

UMAP is an important algorithm that is rapidly gaining popularity as a tool for cytometry data analysis, and we agree that addressing UMAP in the manuscript is appropriate. We have performed UMAP analysis on the datasets presented in the paper. We found that UMAP results are very sensitive to parametrization choices. Therefore, we feel a hasty comparison, without detailed, comprehensive assessment of all of UMAP's adjustable parameters, may unjustly diminish UMAP potential or, on the contrary, be misleadingly optimistic. A careful study that would rigorously examine UMAP parameter selection for various applications would be an important albeit separate manuscript. For this manuscript, we have added a description of our UMAP results in the Discussion of the revised manuscript (lines 444-454 and 477-482) and Suppl. Fig. 8.

- All of the figures are too small. For example, in Figure 1, I appreciate the better separation of the 3,000 iteration plot, but in current image size this corresponds to five pixels. I think that the figures can be restructured into full-page figures (instead of taking only half a page), and that some panels can be reduced (for example, show outputs over less parameter values) and moved to the supplement.

In response to this comment we have adjusted our figures. Specifically, we re-formatted Fig. 1 to dedicate more space to embeddings. For Fig. 2, we reduced the number of output panels and enlarged those that remain. Also, we moved embedding maps from Fig. 3 to Supplementary Fig. 3 and increased their size. We also increased the resolution of the panels we included in the revised manuscript document up to 1 pixel = 1 cell resolution. In addition, we will re-format all figure sizes and resolution and provide production-quality figure files as needed at the proofs/typesetting stage to ensure clear visibility of all components of all the figures.

- For figure 4, I think that there is a mismatch between figure order and text order: 4B is referenced in the text before 4A. Additionally, there is a reference in the text to 4E, which does not exist.

We have edited the revised version of the manuscript to correct this.

- Could the authors please include a table that summarizes their recommendations regarding each of the parameters? The values are spread across the text and it would be beneficial to have a summary either here or in the discussion.

Table 2 was added to the Results section (line 340) to summarize our parameter choice recommendations.

Discussion

- One of the greatest selling points of UMAP is that it provides a better view of the global structure of the data compared to t-SNE. Do the authors think that opt-SNE bridges that gap? Either way, I think that this point deserves mentioning here.

As discussed above, we have added our assessment of UMAP results, how they compared with opt-SNE, and the limitations of our analyses to the discussion section. In response to your question, we believe that t-SNE had a great potential when launched to both represent local structure and provide a fairly good view of the global structure, but this capability declined with an increase in dataset size. Opt-SNE removes the roadblocks that prevent t-SNE to reveal its full capability to visualize datasets of various sizes.

Signed,
El-ad David Amir

We sincerely thank Dr. Amir for his extremely helpful comments and suggestions.

Reviewer #2 (Remarks to the Author):

The article by Belkina et al. describes an algorithm for identification of "optimal" parameters for t-SNE. The article was difficult for us to read as it is presented as a step by step "story" of anecdotal examples and "visual inspections". Unfortunately, despite bold claims, the article does not provide sufficient objective information to support said claims. We have listed some of our major comments below:

1) Starting with the title and name "Opt-tSNE": The claim of a parameter value being optimal is a bold claim that must be taken seriously and supported by evidence. Calling something optimal when evidence at best suggests that it may be an "improvement" would come across as non-scientific, if not misleading. Another example of this is line 264 "... since they provided optimal balance of map quality versus computation time." The theme of making strong statements without clear scientific evidence repeats throughout the manuscript. We have not provided a comprehensive list in this review.

Thank you for pointing this out.

To ensure that accurate terminology is used throughout the paper, we have edited the manuscript as listed below:

1. We have changed the title of the manuscript replacing the word "optimal" with "optimized". In the scientific literature, algorithms that demonstrate improvement in computation time, memory costs, or algorithm-specific evaluation metrics are routinely described as "optimized" (see 3 examples below):

<https://arxiv.org/abs/1706.06786>

An optimized algorithm for multi-scale wideband deconvolution of radio astronomical images.

"...Comparing the two algorithms on a cleaning task of 100,000 iterations, we find that the total wall-clock time to perform 100,000 clean iterations is 223 s in WSCLEAN and 3480 s in CASA. Hence, in this imaging

configuration, the speed of the optimized multi-scale algorithm is over an order of magnitude larger than Cornwell's multi-scale algorithm..."

<https://ieeexplore.ieee.org/document/7724810>

An optimized algorithm for solving travelling salesman problem using greedy cross over operator.

"...It can be evaluated from the table that after 50 iterations the best solution found by proposed greedy operator is better as compared to existing PMX operator. The best solution after 50 iterations is 1983 using proposed greedy operator and 2449 using existing PMX operator. So our proposed operator is 23.5% better as compared to existing PMX operator...."

<https://ieeexplore.ieee.org/document/1524405>

An optimized algorithm of high spatial-temporal efficiency for Megablast

"The optimized algorithm overlaps I/O with computation, further decreases the overall time and the cost of memory, which is only proportional to the size of the database file. The optimized algorithm is suitable to be parallelized on cluster systems...."

Based on our algorithm performance evaluations that are now expanded in the revised manuscript (see lines 310-339), we believe our findings sufficiently fit the definition of "optimized" algorithm.

2. We removed the 'optimal' term from the abstract and from the text where we describe the essence of the opt-SNE toolkit (lines 17, 24, 85, 86, 90, 306, 457; Fig. 1 and Fig. 4 legends).

3. When describing our findings and implications of our results throughout the paper, we have edited the language to make our statements more accurate (lines 24, 83-94, 275, 306, 457, 497).

2) Re. the claim of "dramatic improvement of computation time": There is no comparative study of computation times and the discussion even states that "we have not explicitly focused on efficiency in this work since we benchmarked the algorithm against itself with no specific emphasis on shortening computation time". To validate the claim on computation time improvements, a comprehensive study across several datasets and settings would be required.

We appreciate this comment as we agree with the reviewer that our previous version of our manuscript provided only scarce data/details describing the benchmarking that we performed. We have now included detailed comparisons of the computation time of various embeddings. These data are shown and described in a new section of the Results (starting at the line 310, "**opt-SNE provides faster, higher quality embedding of cytometry datasets**") and the data are shown in Figure 5 and Suppl. Figure 4. In these newly added figures, we provide computation time benchmarks, accuracy scores, and scalability data on the opt-SNE approach.

Especially since on first sight the proposed real-time KLD monitoring approach can potentially result in infinite runtimes (unless there are other control parameters that are not discussed - in which case analysis of those parameters would also be required). This is even further complicated since the results of all these analysis are different and given the unsupervised nature of the data, by definition, there is no way to determine which solution is better and by how much, therefore any runtime comparison would be compromised from the beginning.

Our toolkit already includes a user-accessible hard stop parameter that prevents potential infinite runtimes; since we intend our package for user experimentation with t-SNE parameters, we considered such a hard stop a necessary tool to prevent infinite runtimes. Furthermore, our KLD monitoring stops the embedding optimization when the difference in KLD between the last two iterations becomes less than 0.05% of the KLD absolute value, signaling that there is very little improvement in the KLD value with each new iteration of the embedding. Since KLD minimization drives the t-SNE embedding, we believe that computation times can be compared for t-SNE embeddings with varying parameters where each solution is assessed when the similar KLD minimization rate ($(KLD_{N-1} - KLD_N) < KLD_N/5,000$). Our choice of minimization rate for embedding termination is supported by evaluating accuracy of embeddings terminated at various minimization rates (see Suppl. Figure 4D), but we also kept it as a user-accessible parameter.

3) Re. the claim of "optimal data resolution": There is no quantitative measure of "optimal data resolution" as well as "precise data interpretation". The results claimed are based on visual inspection only and reportedly use "representative examples of multiple runs with varying seed values". These are simply the authors' opinions not scientific evidence backed by empirical data and statistical analysis.

Our response to much of this comment is the same as mentioned above (beginning "To ensure that accurate terminology is used throughout the paper..."). We completely agree with the Reviewer that there is no universally accepted quantitative measure of data resolution quality for t-SNE embeddings. This is a well-known problem (for instance, see Arora et al, 2018, Ref. 43). A commonly used metric with known limitations is 1NN accuracy score that we found to be of limited, but recognized use for our purposes and employed in the classic and more recent publications (van der Maaten's work on bh-tSNE, Becht et al's work on UMAP, Tang et al's LargeVis).

To specifically address the reviewer's comment about seed values, we have now included detailed data on embedding initiated with various seeds into Suppl. Fig. 4C-D. We are aware that a common practical recommendation states that the t-SNE users should test various seeds and choose the embedding with the lowest KLD (see for example van der Maaten's FAQ on t-SNE: <https://lvdmaaten.github.io/tsne/> and his publications on t-SNE). However, it has been widely noted by cytometry data analysts that cytometry datasets produce extremely similar (both visually and KLD-assessed) embeddings when various seeds are used for t-SNE initiation. Even when the run is stopped at a fixed number of iterations and resulting KLD values are slightly different, monitoring KLD beyond the fixed iteration shows that the KLD values that satisfy the $(KLD_{N-1} - KLD_N) < KLD_N/5000$ condition for each seed are very close. Therefore, we are not sure if these data are of much interest to the manuscript readership and would like to leave the decision on keeping them in the manuscript supplementary dataset to the Reviewers and the Editors.

4) Initialization of machine learning algorithms has been subject to research for decades.

That is true. Specifically to t-SNE, most of the research has been targeting the quality of the solution that the algorithm converges to (i.e., how good is the final embedding). Kobak and Berens's PCA-based initialization of the embedding (<https://doi.org/10.1101/453449>) is an example of that, and there have been examples of initializing with results of a previous t-SNE with larger perplexity. Our approach is complementary to what Kobak and others have suggested, and the performance of our method for choosing when to stop early exaggeration and when to end the algorithm is not initialization dependent. On the other hand, the t-SNE convergence time is typically better addressed by adapting the actual algorithm to make it converge faster rather than changing its initialization, which would often lead to inferior quality of the final embedding. Specific examples of addressing the speed of the algorithm are the Barnes-Hut t-SNE introduced by van der Maaten in 2014, and a very recent Flt-SNE adaptation of t-SNE (Linderman et al, Nature Methods 16, 243-245, 2019). Again, opt-SNE is complementary to those approaches and works well together with either BH t-SNE or Flt-SNE. We observe that combining opt-SNE and Flt-SNE is ideal in that opt-SNE provides superior quality embedding on large inputs while Flt-SNE speeds up the convergence to such embedding, which is then further accelerated by opt-SNE guiding Flt-SNE with respect to when to end the early exaggeration phase and when to end the algorithm all together.

It is well known that improved initialization can decrease convergence time. Similarly, a convergence criteria to pause the training process is an idea that has long existed in the field, numerous solutions have been developed, and even some existing tSNE packages already implement. Without objective and quantitative comparison of several state-of-the-art initialization and convergence detection techniques, the authors are simply reporting software development improvements not scientific studies.

We would like to clarify that the first phase (early exaggeration) is not a separate optimization and that we are not proposing to wait until the early exaggeration converges to move onto the next phase. We only wait until the plateau phase is complete before moving on to the next phase, while the typical convergence detection methods would wait much longer. We have now revised our description of the method for clarity on this specific point. Also, after extensive searches, including reaching out to multiple colleagues and experts in the field, we cannot find any publications, other references, or knowledge of non-rudimentary methods for stopping early exaggeration (or the whole optimization) with t-SNE based on convergence evaluation criteria. All t-SNE implementations that we assessed use a fixed number of iterations for the early exaggeration as well as the

total number of iterations to stop the embedding. If the reviewer is aware of a paper or a version of t-SNE that implements some specific convergence criteria, and if they and the editors believe that a comparison with such a method would improve the quality of our manuscript, then we would greatly appreciate being informed of that work.

We feel strongly that this manuscript provides a valuable contribution to the field. Previously, the hyperparameters of tSNE were poorly understood in the context of cytometric data (as evidenced by poor default settings and a lack of published guidance). Our empirical investigation into the effect of each hyperparameter, and the interactions between them, on visualization quality gave rise to our hypothesis that the learning rate should be tuned, and early exaggeration should end when the plateau edge is reached in order to produce high-quality visualizations. We tested this hypothesis on a variety of datasets of different types (flow, mass, and genomic) and sizes (medium to massive) and confirmed, statistically (using 1NN-accuracy) as well as by visual analysis, the improved quality of visualizations obtained. The application of optimized parameters that we propose unlocks the possibility of assessing real life datasets that were completely out of reach for standard t-SNE implementations, and therefore will greatly impact both basic and translational research practices of single cell data analysis.

REVIEWERS' COMMENTS:

Reviewer #1 (Remarks to the Author):

The authors have addressed my suggestions in the manuscript text or in the rebuttal. I have no further comments regarding the manuscript.

-- El-ad David Amir

Reviewer #2 (Remarks to the Author):

The authors have responded to our comments by adding quantitative metrics to demonstrate that opt-SNE provides "faster and higher quality embeddings." This has improved the manuscript. However, taken together, given the lack of careful quantitative and unbiased comparison against a broad set of algorithms and initialization strategies and using a broad range of datasets (see [1]), this article falls short of the stated Aims & Scope of this journal. In the field of single-cell biology in particular, we would note that the impact of visualization-based discovery is limited and the field has largely moved on to quantitative and objective comparison across well-defined groups of samples as opposed to visual inspection.

We would like to reiterate that it is well known that improved initialization can decrease convergence time. Similarly, a convergence criteria to pause the training process is an idea that has long existed in the field, numerous solutions have been developed, and addition of new ones to software packages is relatively simple (example: [2]). From a machine learning perspective, what the authors are reporting is basically software development. Without objective and quantitative comparison of several state-of-the-art initialization and convergence detection techniques, the article's scientific impact in our view will be limited.

[1] Weber, Lukas M., et al. "Essential guidelines for computational method benchmarking." arXiv preprint arXiv:1812.00661 (2018).

APA

[2] <https://jlmelville.github.io/smallvis/optsne.html>